# How Soft Polymers Cope with Cracks and Notches

**Andrea Spagnoli \***[iD]**, Michele Terzano**[iD]**, Roberto Brighenti, Federico Artoni**[iD]
**and Andrea Carpinteri**[iD]

Department of Engineering and Architecture, University of Parma, Parco Area delle Scienze 181/A,
43124 Parma, Italy; michele.terzano@studenti.unipr.it (M.T.); roberto.brighenti@unpr.it (R.B.);
federico.artoni@studenti.unipr.it (F.A.); andrea.carpinteri@unipr.it (A.C.)
**\*** Correspondence: spagnoli@unipr.it; Tel.: +39-0521-905-927

**Abstract:** Soft matter denotes a large category of materials showing unique properties, resulting from a low elastic modulus, a very high deformation capability, time-dependent mechanical behavior, and a peculiar mechanics of damage and fracture. The flaw tolerance, commonly understood as the ability of a given material to withstand external loading in the presence of a defect, is certainly one of the most noticeable attributes. This feature results from a complex and highly entangled microstructure, where the mechanical response to external loading is mainly governed by entropic-related effects. In the present paper, the flaw tolerance of soft elastomeric polymers, subjected to large deformation, is investigated experimentally. In particular, we consider the tensile response of thin plates made of different silicone rubbers, containing defects of various severity at different scales. Full-field strain maps are acquired by means of the Digital Image Correlation (DIC) technique. The experimental results are interpreted by accounting for the blunting of the defects due to large deformation in the material. The effect of blunting is interpreted in terms of reduction of the stress concentration factor generated by the defect, and failure is compared to that of traditional crystalline brittle materials.

**Keywords:** soft materials; polymers; strain rate; defect tolerance; digital image correlation; stress concentrators; notch blunting

## 1. Introduction

A class of materials, which are relevant from the point of view of advanced applications, is represented by the so-called soft or highly deformable materials, such as elastomeric polymers, colloids, liquid-crystals polymers, gels, foams, as well as biological materials—such as soft tissues—that are roughly governed by the same mechanical principles [1]. Typically, these materials have mechanical properties falling within the following range; elastic modulus of 0.1–1.5 MPa, tensile strengths of 1–10 MPa, ultimate tensile strains up to 2000%, and fracture energy of 100–1000 J/m$^2$. Soft materials are endowed with unique features and mechanical properties, explaining the great attention that they have been receiving from the scientific community in the last decades. In natural systems, soft tissues are a fertile source of inspiration for advanced applications, with mechanics and biology going hand-in-hand to formulate the underlying mechanical principles and develop new optimized structural materials [2–6].

The peculiar properties exhibited by soft materials are a direct consequence of their complex and entangled molecular structure, which involves millions or billions of atoms forming linear chains. We owe the fundamentals of the chemical- and physics-based mechanical behavior of this class of materials to the research work of Paul J. Flory and P.J. De Gennes [7–9]. In fully amorphous materials, the structure at the nanoscale level consists of a three-dimensional network of polymer chains, linked together at several discrete points identified as cross-links. The mechanical response of these materials

at the meso- or macroscale is heavily affected by the amount of entanglement and the number of existing cross-links per unit volume, rather than by the bonding strength existing between the atoms, as happens in fully crystalline materials (e.g., metals, ceramics, etc.). Upon the application of load, the deformation induces an alignment and unentanglement of the polymeric chains, with the initial amorphous conformation turned into a semicrystalline-like one. Such a phenomenon usually occurs at quite large deformation, thanks to the highly-oriented arrangement of the chains along the tensile direction [10–12]. This so-called strain-induced crystallization is responsible for the typical stiffening behavior which is noticed in the stress–strain curve of polymers at high strain levels [13,14].

Soft materials normally can deform several times their original length, without being damaged or ruptured. If macroscopic flaws are introduced in the form of cuts, cracks or notches, the stretchability can be affected in different measure, depending on the flaw sensitivity of the material. The concept of a material length scale, separating flaw-sensitive from flaw-insensitive rupture, has been initially proposed for hard materials [15] and later extended to soft polymers [16]. Differently from traditional crystalline solids, rupture of soft polymers occurs at large deformation, when the rearrangement of the polymeric chains leads to flaw reshaping and strengthening around the highest strained region. In particular, an existing initial sharp crack blunts significantly before propagation, with the effect of relieving the strain concentration around the tip [17–19]. The resulting feature is an enhanced defect tolerance, with some tough soft materials showing an insensitivity to flaws up to a few millimeters long, in contrast to the typical nanometer scale for hard brittle solids [20]. For instance, biological tissues such as skin are known for their extreme resistance to flaws, which makes virtually impossible to propagate tearing in a stretched sample. Experimental observations on rabbit skin showed that a small notch does not propagate but progressively blunts, due to straightening and stretching of the collagen fibers [21]. However, other soft materials, such as the so-called hydrogels, are often prone to premature fracture and have low fatigue resistance. In such materials, the rupture process is strongly influenced by the fluid interaction [22], and it has been discovered that increased toughness and fatigue resistance are obtained through the development of polymer networks containing chains of different lengths (such as the double-network hydrogels, containing both a short- and a long-chain highly stretchable network). In this fashion, the shortest chains act as sacrificial elements, while the longer ones provide the material with a further elastic behavior [23,24]. From this perspective, it appears that the polymer network characteristics play a crucial role in defining the macroscopic behavior of the material. Furthermore, soft materials are sensible to time-dependent effects, with the fracture energy depending on the rate of application of the external loads, because of the viscous energy dissipation occurring in the crack tip region [25–27]. Sometimes, under a constant applied load, a so-called delayed fracture has been observed, depending on the network structure of the soft matter [28].

In this work we present a comprehensive investigation into the defect tolerance of flawed specimens of rubber-like polymers, with a detailed summary of experimental findings recently published by the authors [29–31]. Various configurations with cracks and notches are examined in order to evaluate the macroscopic mechanical response in relation to the flaw severity. Moreover, the effect of the applied strain rate is taken into account phenomenologically. All the experimental tests are conducted under strain control, and the kinematically-related quantities are measured through a contactless Digital Image Correlation (DIC) technique. Due to the local high deformations in proximity of the defects, a severe defect remodeling with evident blunting is noticed. The main purpose is to show how crack blunting affects the rupture behavior of soft materials. Through a simplified analytical model, the increase in the curvature radius at the notch root with the remote applied load is described. Such a model is applied to the experimental results, putting into evidence that it is the blunting effect which controls the rupture process; in particular, the grade of blunting appears to determine the transition from the typical small-scale yielding failure to rupture at a constant theoretical strength.

The paper is organized as follows. Section 2 presents a collection of the results obtained from the experiments, with details on the materials and methods employed. Accurate images taken from the DIC elaboration are included. In Section 3, a detailed discussion is developed in order to give a

comprehensive interpretation of the experiments, with specific reference to the notch blunting effect, which is computed through a simplified analytical model. Finally, Section 4 sets out the conclusions.

## 2. Experimental Tests on Macroscopically Flawed Thin Plates

In this section, we present the results of tensile tests on thin plates made of various elastomeric polymers, carried out under displacement control up to complete failure. The plates contain flaws with different geometrical features, ranging from cracks to notches, denoted here as geometric discontinuities with a finite radius of curvature. Rubber-like polymers have been chosen because their behavior is highly representative of a vast range of soft materials. For instance, silicone rubber is often employed as a substitute of human skin, since it does not show the strain-induced stiffening of natural rubbers, maintaining comparable values of tensile and tearing strength [32]. The stress–strain curves obtained from tensile testing of silicone rubbers are typical of a hyperelastic, almost incompressible behavior. Several models have been proposed [33], which usually show good agreement at low to moderate stretches (generally below 1.5). During our experiments we have found that the one-parameter neo-Hookean model offers a satisfactory approximation in the stretch range of interest (see, for instance, Figure 1).

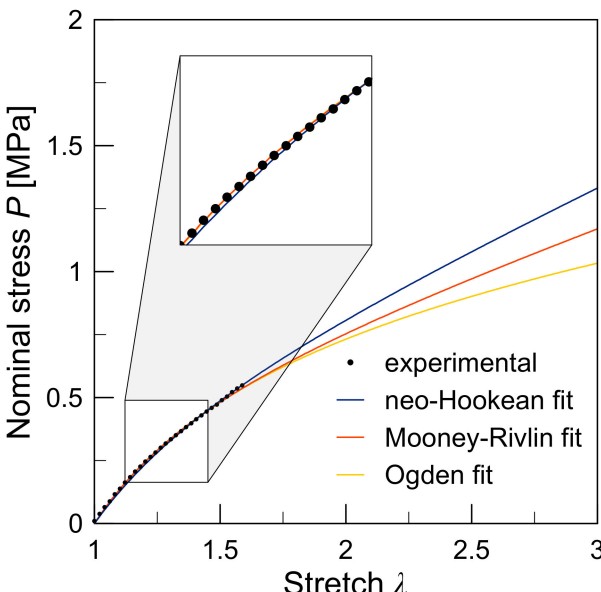

**Figure 1.** Typical curve obtained from tensile testing of a silicone rubber (experimental and fitted data). The enlarged inlet shows the region λ = 1.2–1.4, with excellent agreement between the different models.

The purpose of the experiments herein described is to evaluate the effect of initial flaws on the tensile strength of the material, considering both the effect of the flaw size and of the applied strain rate. The response of the specimens during the experimental tests is monitored by measuring the applied force and corresponding displacements, and by using the Digital Image Correlation (DIC) technique. The DIC is a contactless technique widely used to get full-field displacement and strain maps in experiments, through numerical reconstruction of the kinematic field shown by the surface points of the samples. For an optimal use of such a technique, the surface of the specimens needs to be covered with speckle patterns before testing. Several parameters affect measurement accuracy and spatial resolution, included optical measures connected to the camera and lens resolution, image magnification, mean size of the speckle pattern, and factors depending on the correlation algorithm, such as the image subset size and the gray-level interpolation. In particular, the resolution of the displacement measurement is governed by the subset size, whose lower bound is limited by the speckle size and, consequently, by the available pixels [34,35]. In the present work, the images are acquired with a high-resolution digital camera (maximum resolution of 5184 × 3456 pixels) mounted

on a tripod, and lights are used to ensure a uniform illumination of the samples. The sequence of images is treated by means of the freeware software Ncorr [36], developed in MATLAB environment, for monitoring the displacement and the strain fields within the specimen.

### 2.1. Plates Containing Elliptical Flaws

The first set of experiments is carried out on elastomeric sheets under tension, containing elliptical flaws of length $2a$, characterized by a finite notch root radius (Figure 2a). The plates are made of Sylgard®, a common silicone polymer having the following elastic parameters: initial Young modulus $E = 0.84$ MPa and Poisson ratio $\nu = 0.37$. Three types of specimens have been prepared, containing a centered elliptical flaw with different values of the root radius $\rho$ and same length $2a$. The plate aspect ratio is kept constant at $L/W \approx 2$. The geometric characteristics of the specimens are reported in Table 1. The flawed plates are subjected to tensile loading along the $y$-axis, applied at a constant strain rate of $\dot{\varepsilon}_0 = \dot{\delta}/(2L) = 4.8 \cdot 10^{-3} \ s^{-1}$. The tests have been interrupted before failure, since an evident notch remodeling was noticed. The response during the experimental tests is monitored by means of the DIC technique.

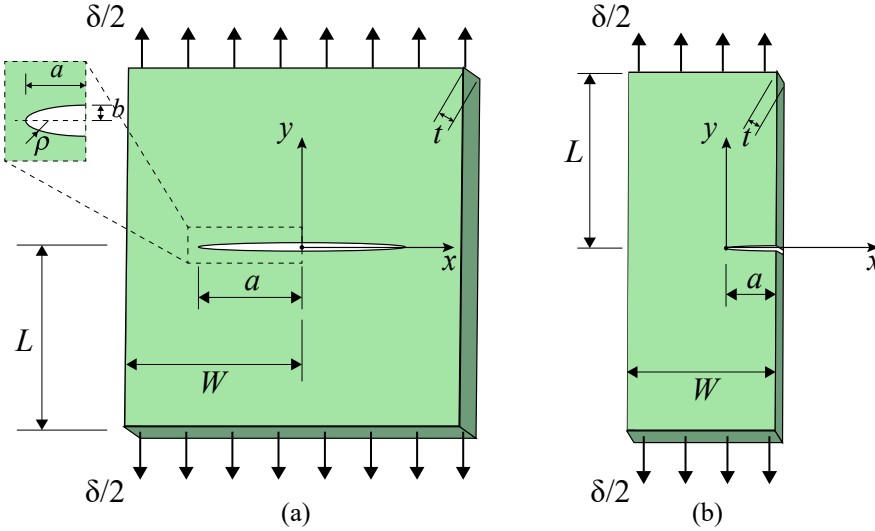

**Figure 2.** (**a**) Sketch of a thin plate containing a centered internal flaw. The enlarged view shows the case of an elliptical notch, with semi-axes $a$ and $b$. $\rho$ is the tip radius of the notch ($\rho = 0$ in the case of a crack-like flaw). (**b**) Sketch of an edge-cracked plate. The plates are subjected to a remote displacement $\delta$ applied along the $y$-axis.

**Table 1.** Geometric characteristics of the thin plates with elliptical notches. The rightmost column contains the values of the initial notch root radius $\rho = b^2/a$.

| Specimen ID | W (mm) | a (mm) | b (mm) | t (mm) | a/W (-) | ρ (mm) |
|:---:|:---:|:---:|:---:|:---:|:---:|:---:|
| El1 | 58.5 | 20 | 5 | 2.0 | 0.342 | 1.250 |
| El2 | 58.5 | 20 | 2.5 | 2.0 | 0.342 | 0.3125 |
| El3 | 58.5 | 20 | 1 | 2.0 | 0.342 | 0.0050 |

Figure 3 shows the initial (undeformed) and the generic stretched shapes of the specimen El1 at two increasing levels of the remote applied stretch. The corresponding strain patterns obtained from the DIC analysis are illustrated in Figure 4, specifically, the Green–Lagrange strain $E_{yy}$ parallel to the

loading direction and the strain $E_{xx}$ transversal to such a direction. Three different levels of the applied remote stretch are considered, defined as

$$\lambda_0 = 1 + \frac{\delta}{2L} \qquad (1)$$

where $\delta$ is the applied displacement.

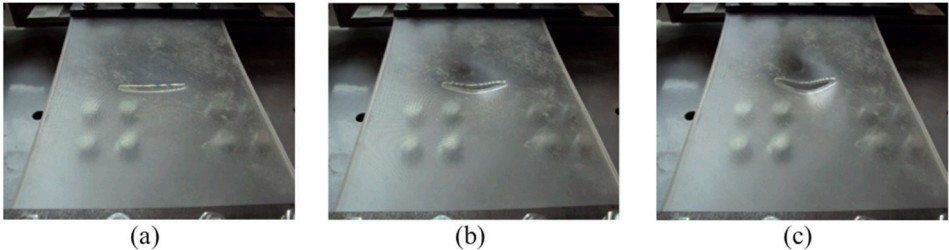

(a)　　　　　　　(b)　　　　　　　(c)

**Figure 3.** Qualitative view of the (**a**) undeformed and (**b**,**c**) increasingly deformed configurations, in the flawed specimen El1.

The DIC plots show that the material is able to comply with very high deformations, leading to a severe defect remodeling characterized by an evident blunting of the notch. A compressed region just in front and behind the elliptical hole is observed, due to the contraction effect arising in the direction normal to the applied load (see also Figure 3c). This phenomenon has a beneficial effect in terms of the strain concentration, with a sort of augmented notch blunting due to the local flexural instability of the thin plates in the compressed zones [29].

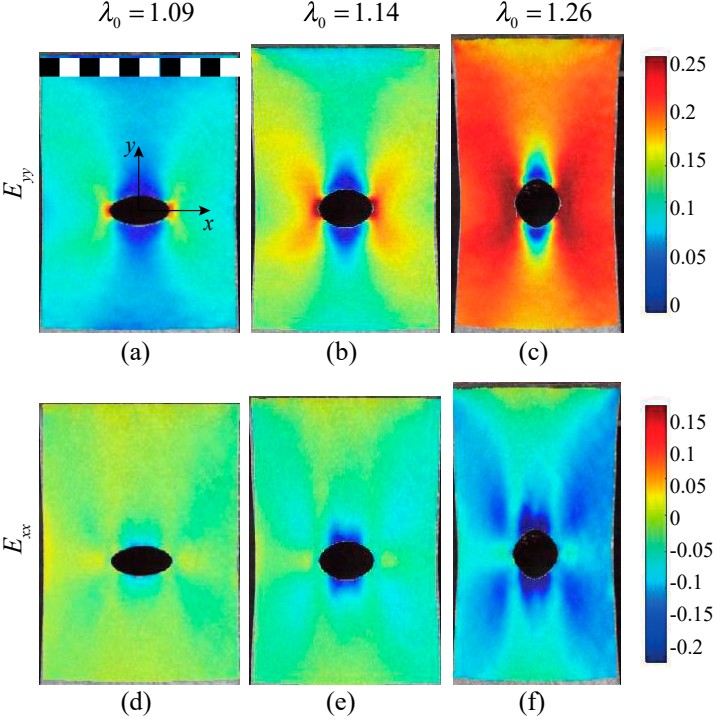

**Figure 4.** Strain maps obtained from the Digital Image Correlation (DIC) analyses on the elliptically notched specimen El1, for three increasing values of the applied remote stretch $\lambda_0$. (**a**–**c**) Green–Lagrange strain $E_{yy}$; (**d**–**f**) corresponding maps for the strains $E_{xx}$. In (**a**), the length scale is expressed in cm.

## 2.2. Plates Containing Internal Crack-Like Flaws

A second set of experiments deals with plates containing a centered crack of length 2*a* (Figure 2a with $b = \rho = 0$). The samples are made of a commercial silicone rubber (TSE3478T by Momentive). The desired crack is obtained by manually cutting the samples with a sharp blade. From tensile tests on sound specimens, the initial Young modulus of the material is found approximately equal to $E = 1.12$ MPa and the Poisson ratio to $\nu = 0.42$. Four series of specimens containing a centered crack are here considered, characterized by different values of the relative crack length *a/W*, whereas the plate aspect ratio is kept constant at $L/W \approx 2$. The geometric characteristics of the specimens are reported in Table 2. The plates are subjected to tensile loading along the *y*-axis up to failure, applied at a constant strain rate of $\dot{\varepsilon}_0 = 5.8 \cdot 10^{-3}$ s$^{-1}$. The response during the experimental tests is monitored by means of the DIC technique.

**Table 2.** Geometric characteristics of the specimens containing a centered crack. The rightmost column contains the ultimate remote stretch before failure.

| Specimen ID | W (mm) | a (mm) | t (mm) | a/W | $\lambda_U$ |
|:-----------:|:------:|:------:|:------:|:-----:|:-----------:|
| CC1 | 56 | 10 | 2.75 | 0.179 | 1.75 |
| CC2 | 56 | 15 | 3.00 | 0.268 | 1.61 |
| CC3 | 56 | 20 | 2.75 | 0.357 | 1.55 |
| CC4 | 56 | 25 | 2.85 | 0.446 | 1.32 |

Figure 5 shows the deformed patterns of the specimens, for three different stages of the applied loading: at the beginning of the test (undeformed, left column), at an intermediate stage (remote stretch $\lambda_0 = 1.29$, central column), and at the final stage before failure (ultimate remote stretch, right column). The strain patterns obtained from the DIC analyses are also shown, specifically, the Green–Lagrange strain $E_{yy}$ (parallel to the loading direction) at the intermediate and the ultimate stages. It can be noticed that the initial crack-like shape tends to blunt under loading and the applied ultimate stretch generally decreases for an increasing relative crack size *a/W*. At the intermediate stage, the DIC maps clearly show a strain concentration typical of an elliptical notch, where the maximum strain values occur in the locations corresponding to the original crack tips. At incipient failure, the strain maps exhibit a complex distribution due to the failure mechanisms developing in the vicinity of the notch roots. It is worth noticing that out-of-plane displacements occur in two limited regions close to the crack edges (the blue regions in Figure 5), because of the contraction effect arising in the direction normal to the applied load, i.e., in the *x*-direction.

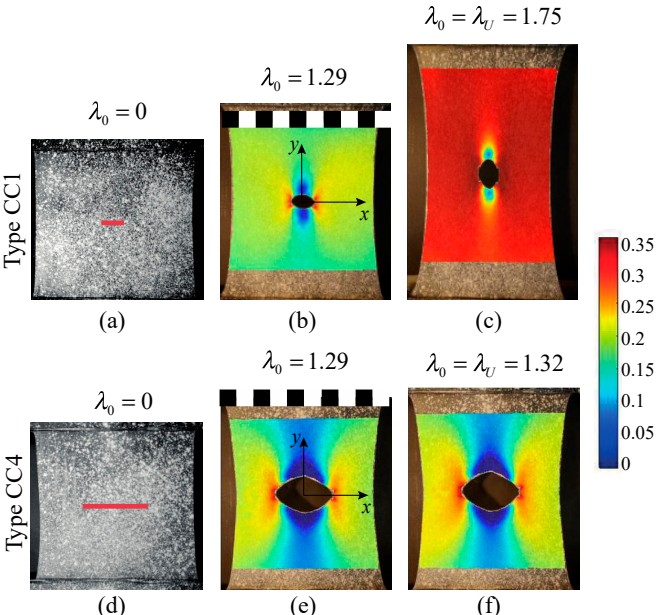

**Figure 5.** Images of the precracked specimens at the initial stage (left column), and with maps of the Green–Lagrange strain $E_{yy}$ at an intermediate stage (central column) and final stage at incipient failure (right column). The remote stretch is shown on the top of each plot. (**a**–**c**) Specimens type CC1 ($a/W = 0.179$); (**d**–**f**) specimens type CC4 ($a/W = 0.446$). In (b,e), the length scale is expressed in cm.

The distribution of the Green–Lagrange strain $E_{yy}$ along the *x*-axis is shown in Figure 6.

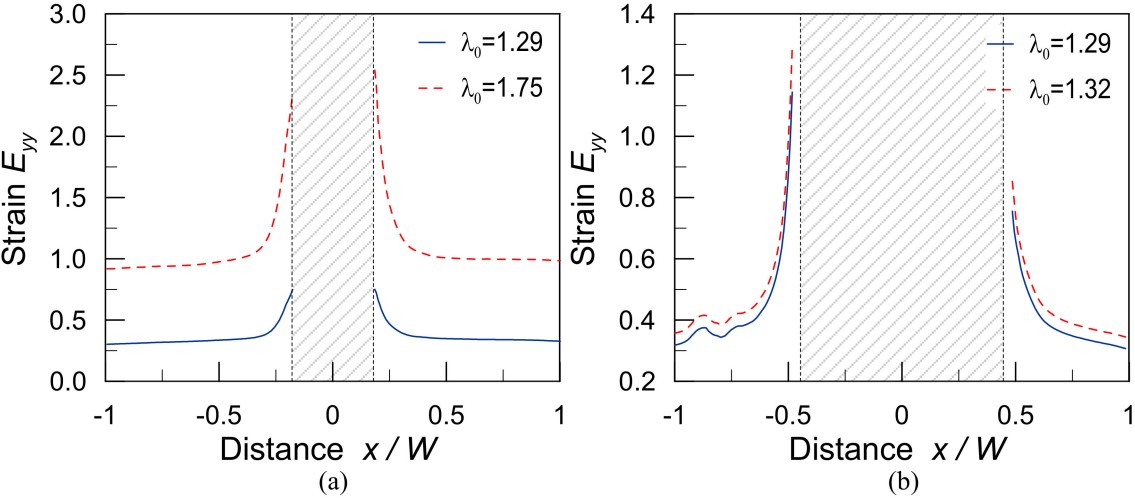

**Figure 6.** Green–Lagrange strain $E_{yy}$ distribution along the cracked section of the plate, at an intermediate (continuous lines) and ultimate values (dashed lines) of the remote applied stretch. (**a**) Specimens type CC1 ($a/W = 0.179$) and (**b**) specimens type CC4 ($a/W = 0.446$).

### 2.3. Plates Containing Edge Crack-Like Flaws

The last set of experiments that we review is related to plates containing edge crack-like defects (Figure 2b), where we consider the effects of the relative crack size $a/W$ and of the applied strain rate. A first group of edge-cracked plates (specimens EC1–EC4 in Table 3) has been prepared with the commercial silicone RTV 615 (Elantas Europe Srl). From tensile tests on sound specimens, the initial Young modulus of the material is equal to $E = 1.50$ MPa and the Poisson ratio is equal to $\nu = 0.42$. The plates are subjected to tensile loading along the *y*-axis up to failure, applied with three different

strain rates: $\dot{\varepsilon}_0^{(1)} = 1.9 \cdot 10^{-3}$ s$^{-1}$, $\dot{\varepsilon}_0^{(2)} = 0.48 \cdot 10^{-3}$ s$^{-1}$, and $\dot{\varepsilon}_0^{(3)} = 0.16 \cdot 10^{-3}$ s$^{-1}$. The response during the experimental tests is monitored by means of the DIC technique.

**Table 3.** Geometric characteristics of the edge-cracked plates. The rightmost column contains the ultimate remote stretch before failure. In specimens EC1–EC4, the fastest rate $\dot{\varepsilon}_0^{(1)} = 1.9 \cdot 10^{-3}$ s$^{-1}$ is considered.

| Specimen ID | $W$ (mm) | $a$ (mm) | $t$ (mm) | $a/W$ | $\lambda_U$ |
|:---:|:---:|:---:|:---:|:---:|:---:|
| EC1 | 26.3 | 1 | 2.3–3.5 | 0.038 | 1.26 |
| EC2 | 26 | 2 | 2.3–3.2 | 0.077 | 1.18 |
| EC3 | 26 | 5 | 2.3–2.9 | 0.192 | 1.11 |
| EC4 | 26 | 8 | 2.5–2.9 | 0.308 | 1.07 |
| EC5 | 25 | 1.8 | 4.2 | 0.072 | 1.42 |
| EC6 | 25.5 | 3 | 4.2 | 0.118 | 1.36 |
| EC7 | 26 | 4 | 4.3 | 0.154 | 1.28 |
| EC8 | 25 | 5 | 4.6 | 0.202 | 1.26 |

A second group of edge-cracked plates (EC5-EC8 in Table 3) has been prepared using a different silicone rubber (Elite Double 32 by Zhermack Dental), with an estimated Young modulus equal to $E = 1.36$ MPa and the Poisson ratio is equal to $\nu = 0.42$. The plates are subjected to tensile loading along the $y$-axis up to failure; such a loading is applied with a strain rate $\dot{\varepsilon}_0 = 3.8 - 4.2 \cdot 10^{-3}$ s$^{-1}$. Four series of specimens have been prepared for each group, with different values of the relative crack length $a/W$, while the plate aspect ratio is kept constant at $L/W \approx 1.5$.

The results obtained from the DIC analyses in the specimens EC2–EC4 are illustrated in Figure 7, specifically, the Green–Lagrange strain $E_{yy}$ (parallel to the loading direction) at incipient failure for the highest strain rate among the three considered. From a qualitative observation, it can be noticed that crack tip blunting is limited (compared, for instance, to the case of an internal crack in Figure 5).

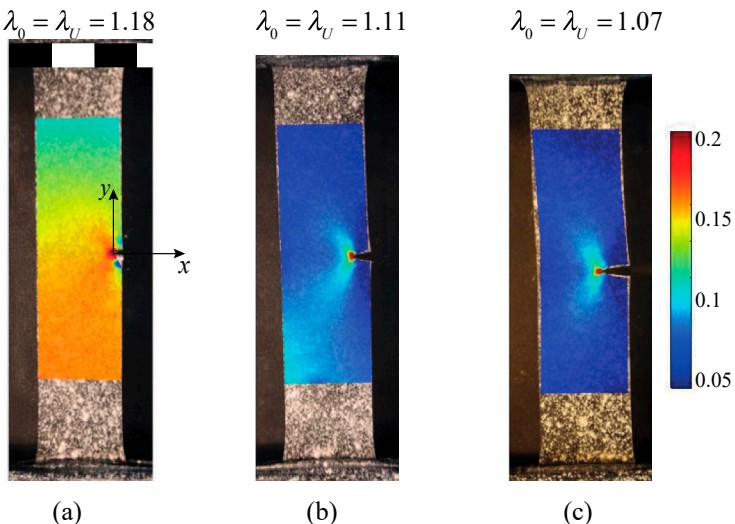

**Figure 7.** Map of the Green–Lagrange strain $E_{yy}$ at incipient failure, at the applied rate $\dot{\varepsilon}_0^{(1)}$. The remote stretch is shown on the top of each plot. (**a**) Specimens type EC2 ($a/W = 0.077$), (**b**) type EC3 ($a/W = 0.192$) and (**c**) type EC4 ($a/W = 0.308$). In (**a**), the length scale is expressed in cm.

The results show that specimens with only one ligament zone are more sensible to defects with respect to the ones containing a central crack (CC1–CC4). The eccentric load effect, which increases

with the deformation, plays a crucial role in intensifying the stress close to the crack tip, leading to a premature failure of the polymer network chains within the crack process region.

### 2.4. Effects of Intrinsic Material Defects

The group of experiments on edge-cracked plates has also involved samples with intrinsic defects, in the form of microvoids (D1–D4 in Table 4). For this purpose, we have prepared samples with the same material and sizes of the group EC1–EC4, but following a different treatment during the preparation of the silicone mixture. To prepare the samples, 50 g of component A (matrix) are thoroughly mixed with 5 g of component B (curing agent). At this point, the mixture for the material without defects is carefully degassed in vacuo and subsequently mechanically spread into the custom-made aluminum mold. This stage is followed by a second degassing, then the mixture is cured in oven at 60 °C overnight and finally mechanically removed from the mold. In order to obtain microbubbles embedded in the material, the silicone mixture is directly cured in oven without the degassing stages.

**Table 4.** Geometric characteristics of the edge-cracked plates with intrinsic defects. The rightmost column contains the ultimate remote stretch before failure.

| Specimen ID | $W$ (mm) | $a$ (mm) | $t$ (mm) | $a/W$ | $\lambda_U$ |
|:-----------:|:--------:|:--------:|:--------:|:-----:|:-----------:|
| D1 | 26.3 | 1 | 2.7–3.3 | 0.038 | 1.23 |
| D2 | 26 | 2 | 2.7–3.1 | 0.077 | 1.15 |
| D3 | 26 | 5 | 2.8–3.0 | 0.192 | 1.12 |
| D4 | 26 | 8 | 2.8–3.5 | 0.308 | 1.09 |

A comparison of the materials, with and without the microvoids, is shown in Figure 8a,b. The average void radius is equal to $r_v = 0.35$ mm, and its distribution is described by adopting a normal probability function (variance equal to 0.021 mm$^2$). The plot of the probability function (Figure 8c) shows a good agreement with the measured distribution, and ~80% of the voids have a radius within half of the smallest edge crack length considered.

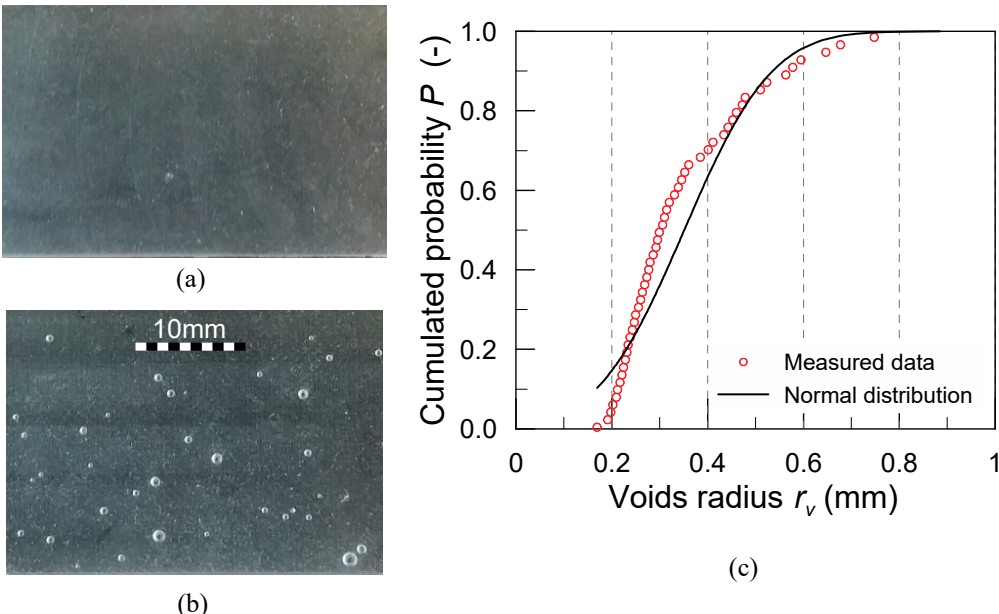

**Figure 8.** Images of the elastomeric material employed in specimens EC1–EC4 and D1–D4. Detail of the material (**a**) without and (**b**) with intrinsic defects. (**c**) Distribution of the void size cumulated probability.

Figure 9 illustrates the failure behavior of edge-cracked specimens EC1–EC4 and D1–D4, in terms of the ultimate stretch vs the relative crack size. As was expected, smaller ultimate stretches are attained for increasing values of the relative crack size *a/W*. The decreasing trend, with a roughly quadratic pattern, confirms that the material is sensitive to the presence of the initial flaws.

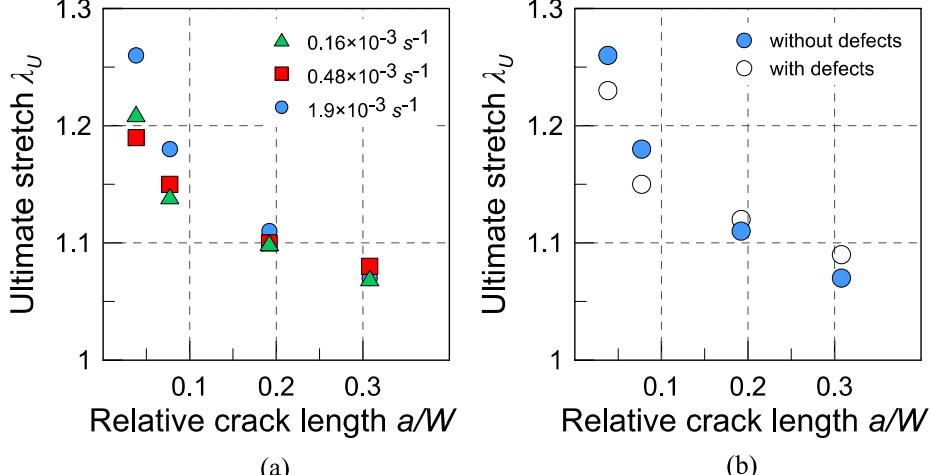

**Figure 9.** Ultimate stretch vs the initial relative crack size for the edge-cracked samples. (**a**) Effect of the applied strain rate in the specimens EC1–EC4 and (**b**) influence of the material defects, for a strain rate $\dot{\varepsilon}_0 = 1.9 \cdot 10^{-3}\ \mathrm{s}^{-1}$ (results from samples EC1–EC4 are shown as solid circles, and D1–D4 as hollow circles).

The effect of the strain rate is investigated in Figure 9a. The highest strain rate provides the highest ultimate stretch, although the effect tends to reduce at lower ultimate stretches, suggesting that the influence of flaw size on the ultimate stretch decreases if the deformation is applied sufficiently slow. Figure 9b compares the failure in specimens with and without intrinsic material defects. It can be inferred that microdefects tend to anticipate failure, although the results are not so clear, probably due to the uneven distribution of the microbubbles in the samples.

## 3. Discussion

### 3.1. Model for Notch Blunting

Cracks and notches in soft materials become distorted under loading due to large strain effects, with an evident tip blunting, as is clearly shown in the figures obtained from the DIC on the experimental tests. In order to quantify the flaw severity and account for the geometrical effect of blunting, we resort to the concept of the stress concentration factor. Considering a notched sample under remote uniform stress acting parallel to the *y*-axis, the stress concentration factor can be defined as

$$K_t = \sigma_{\max} / \sigma_0 \tag{2}$$

where $\sigma_{\max} = \sigma_{yy,\max}$ is the maximum notch root stress and $\sigma_0$ is the corresponding remote stress, measured with respect to a uniformly stressed (gross) section of the sample. In the following, we propose a simplified analytical model, capable of relating the stress concentration factor with the tip radius $\rho$ of the blunted notch and explore the variation of $K_t(\rho)$ under increasing applied loading. Such an approach, conceived for the analysis of blunting of elliptical notches [29], is extended to the tip blunting of cracks (Section 3.2).

The starting point is the solution for an elliptical notch in an infinite elastic plate, having semi-axes $a$ and $b$. Its equation can be written as follows

$$y(x) = \frac{b}{a}\sqrt{a^2 - x^2} \tag{3}$$

and the radius of curvature at the tips ($x = \pm a$) is equal to

$$\rho = b^2/a \tag{4}$$

The stress concentration factor $K_t$ for an elliptical notch is obtained from the renowned Inglis' solution [37]:

$$K_t(\rho) = \left(1 + 2\sqrt{\frac{a}{\rho}}\right)Y\left(\frac{a}{W}, \frac{a}{b}\right) \tag{5}$$

where $Y$ is a corrective factor introduced to account for the finite width of the specimens [38] (considering the initial geometry of the tested notched specimens, the values of $Y$ range from 1.09 to 1.10). For the sake of simplicity, this factor is taken equal to unity in the following equations.

In order to describe the change of the stress concentration factor with the increasing deformation of the blunted tip under loading, we need to consider the variation of the radius of curvature. Such a variation depends on the deformation state in proximity of the notch root, which has to be evaluated in the deformed configuration (see Figure 10a). Let us consider a small square element of edge length equal to $h$, located in the proximity of the notch root, with edges inclined by an angle $\alpha$ with respect to the horizontal axis (Figure 10b). The small strain tensor $\varepsilon'$ in the local reference system 1–2 (with components $\varepsilon_{11}, \varepsilon_{22}, \varepsilon_{12}$) is related to the corresponding tensor $\varepsilon$ (with components $\varepsilon_{tt}, \varepsilon_{nn}, \varepsilon_{tn}$) in the reference system $t$-$n$, with its origin located at the notch root, through the well-known relationships [33]:

$$\varepsilon_{11} = c^2\varepsilon_{tt} + s^2\varepsilon_{nn}, \varepsilon_{22} = s^2\varepsilon_{tt} + c^2\varepsilon_{nn}, \varepsilon_{12} = cs\varepsilon_{tt} - cs\varepsilon_{nn} \tag{6}$$

with $c = \cos\alpha, s = \sin\alpha$. Assuming plane stress conditions and the material governed by the generalized Hooke model, the strain tensor components are

$$\varepsilon_{tt} = -\nu\sigma_{nn}/E, \varepsilon_{nn} = \sigma_{nn}/E, \varepsilon_{tn} = 0 \tag{7}$$

If the angle $\alpha$ is sufficiently small, the only non-zero component of the stress tensor is

$$\sigma_{nn} = K_t(\rho)\sigma_0 \tag{8}$$

The radius of curvature at the notch root, in the undeformed state (point A in Figure 10a,b), is approximated by the radius of the local osculating circle:

$$\rho \simeq \frac{h}{2\cos\beta} \tag{9}$$

with $\beta = \pi/2 - \alpha$, and the increased radius of curvature in the deformed state (point A' in Figure 10a,b) is approximated as

$$\rho' = \frac{h(1 + \varepsilon_{22})}{2\cos(\beta + \gamma)} \tag{10}$$

where $\gamma = 2\varepsilon_{12}$.

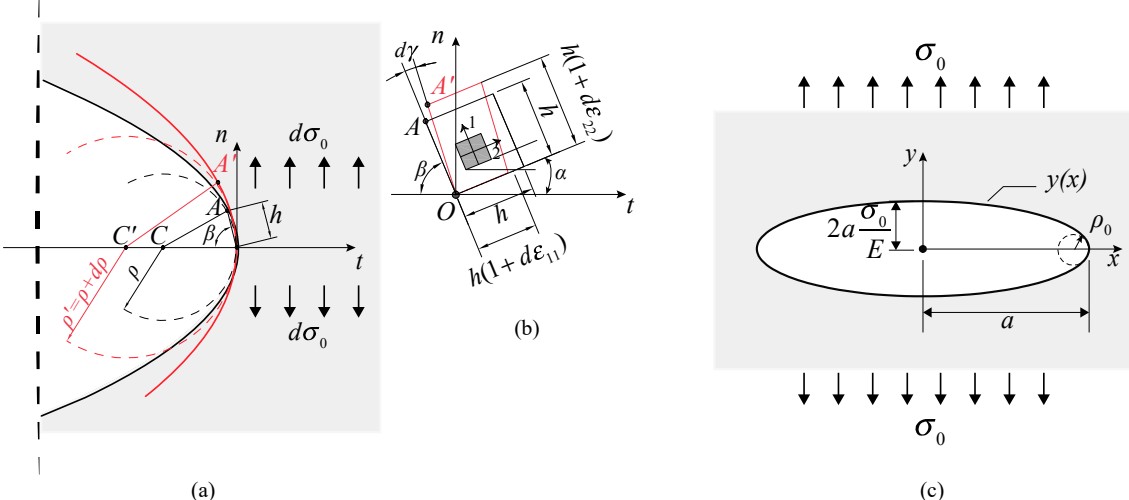

**Figure 10.** (**a**) Model of notch blunting. Schematic of the notch profile modification due to the large strains occurring in the material during an increment of the remote stress $d\sigma_0$. The radius of curvature at point $A$ (undeformed) and $A'$ (deformed) is shown. (**b**) Deformation of a small material element in the vicinity of the notch root. (**c**) Elliptical notch, with the equivalent semi-axes obtained from the crack flank displacement under tensile stress $\sigma_0$.

The model here described is nonlinear, because the local strains depend on $K_t$ through Equations (6)–(8), where, in turn, $\rho$ is also a function of the local strains through Equation (10). Following an incremental procedure, firstly the increments of strain are evaluated for an applied remote stress variation $d\sigma_0$ through Equations (6)–(8); then, the updated radius of curvature is obtained from Equation (10) and the stress concentration factor is computed from Equation (5). In other words, following a stepwise updated Lagrangian approach, at the first increment the increased radius of curvature $\rho'$ of Equation (10) is calculated with respect to the reference configuration on the undeformed notch, whilst at successive increments the reference configuration, i.e., $\rho$ from Equation (9), corresponds to the radius of the notch root under the current stress level. The results applied to one of the elliptically-notched plates tested experimentally are illustrated in Figure 11.

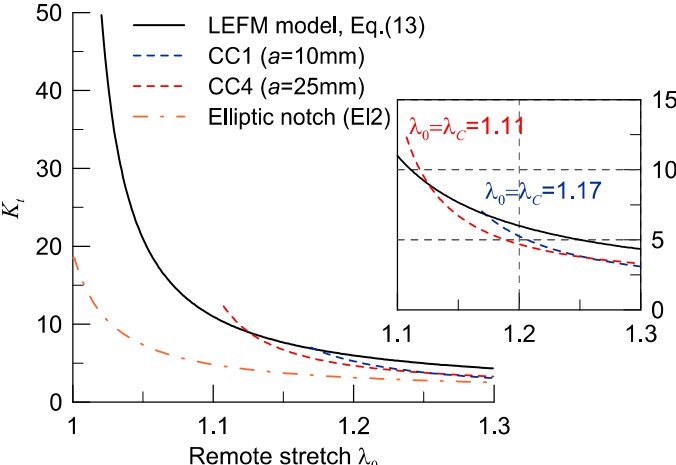

**Figure 11.** Stress concentration factor as a function of the remote applied stretch. Prediction of the 'pure' LEFM model (continuous line) and results obtained from the 'modified' model, in the center-cracked specimens CC1 and CC4. The dash-dot line refers to the elliptically-notched plate El2. The enlarged view shows the region $1.1 < \lambda < 1.3$.

### 3.2. Crack Tip Blunting

Linear Elastic Fracture Mechanics (LEFM) is grounded on the hypothesis of infinitesimal displacements, with the near-tip region being described by the K-dominated fields, which imply, for instance, a square root singularity in the strain and stress components at the crack tip. In soft materials, the large strain region near the tip can be relevant, and the local fields need to be defined within the framework of finite strain elastostatic, showing for instance stress singularities different from the well-known square-root [39]. However, tip blunting can still be described according to the simplified model introduced in Section 3.1, provided that the local strain is not too large.

According to LEFM, the deformed shape of a generic crack of semilength $a$, subjected to a uniform stress $\sigma_0$, is described by the following expression of the crack flank displacement [40].

$$y(x) = 2\frac{\sigma_0}{E}\sqrt{a^2 - x^2} \tag{11}$$

which is the equation of an ellipse, having the major semi-axis equal to $a$ and the minor semi-axis equal to $b_0 = 2a\frac{\sigma_0}{E}$ (Figure 10c). Retrieving the expression of the tip radius of an ellipse from Equation (4), the equivalent radius of curvature is

$$\rho_0 = 4a\left(\frac{\sigma_0}{E}\right)^2 \tag{12}$$

A direct relationship between the applied remote stretch and the stress concentration factor predicted by the Inglis solution, Equation (5), is derived as

$$K_t(\lambda_0) = 1 + 2\sqrt{\frac{a}{\rho_0}} = \frac{\lambda_0}{\lambda_0 - 1} \tag{13}$$

where we have used the standard relation for linear elastic materials: $\lambda_0 = 1 + \frac{\sigma_0}{E}$. The noticeable feature of such an expression is that it does not depend on the initial length of the crack. According to LEFM, the ultimate condition at failure occurs when the remote stress equals

$$\sigma_C = \frac{1}{Z\left(\frac{a}{W}\right)}\sqrt{\frac{G_C E}{\pi a}} \tag{14}$$

where $Z\left(\frac{a}{W}\right)$ is a corrective factor for the finite width of the cracked specimens [41] (considering the initial geometry of the tested notched specimens, the values of $Z$ range from 1.02 to 1.69) and $G_C$ is the fracture resistance of the material. The critical stretch, omitting the corrective factor $Z$, is then obtained as

$$\lambda_C = 1 + \sqrt{\frac{G_C}{\pi E}} a^{-\frac{1}{2}} \tag{15}$$

Up to this point, we have not considered the deformation of the blunted crack under loading. The predicted stress concentration factor, as is described from this 'pure' LEFM model in Equation (13), is illustrated by the continuous line in Figure 11.

The effects of the blunted tip deformation under increasing loading and the resulting stress concentration factor are described adopting the 'modified' model introduced in Section 3.1, where the current tip radius is computed on the deformed configuration from Equation (10). We assume that the initial configuration of the crack is the critical condition of LEFM, Equation (14), when $\sigma_0 = \sigma_C$ and the corresponding value of the radius of curvature, from Equation (12), is

$$\rho_C = \frac{4}{\pi}\frac{G_C}{E} \tag{16}$$

The results of the 'modified' model are also shown in Figure 11. We have considered the center-cracked specimens CC1 and CC4, which present two different lengths of the crack: we notice that, contrary to the LEFM model results, there is a dependence of $K_t$ on $a$.

### 3.3. Application to Experimental Data

The model is applied to account for the effect of blunting in the failure of the specimens tested during the experiments. The data collected in Figure 12 illustrate the ultimate condition, when the remote stretch is equal to $\lambda_0 = \lambda_U$, with respect to the initial length $a$ of the defect. In this plot we have also added the predicted trend of LEFM, as obtained from Equation (15), where an average value of the fracture resistance for silicone rubbers is taken equal to $G_c = 1 \text{ kJ/m}^2$.

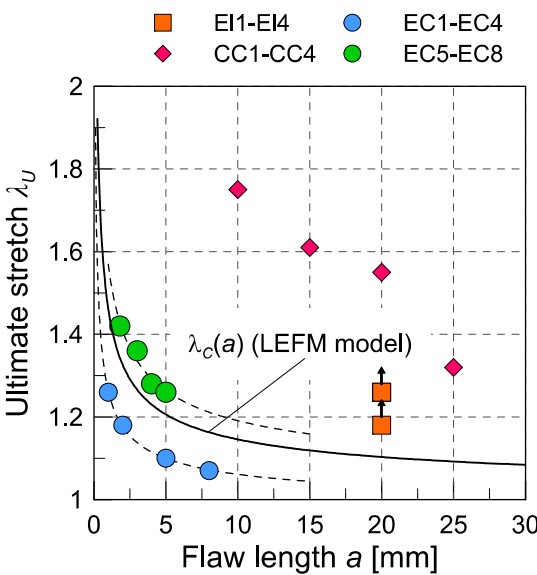

**Figure 12.** Ultimate remote stretches at failure $\lambda_U$ as a function of the initial crack length $a$. In the case of the elliptical notches, the upward arrows indicate that failure has not occurred within the considered range of applied stretch. The continuous line corresponds to the prediction of the 'pure' LEFM solution, Equation (15). The dashed lines correspond to best-fit curves with a $\lambda_U$ vs. $a$ power law dependence with exponent $-0.5$. Results for EC1–EC4 for a strain rate $\dot{\varepsilon}_0 = 1.9 \cdot 10^{-3} \text{ s}^{-1}$.

It should be noticed that the plot in Figure 12 is by no means a representation of the flaw sensitivity in the canonical form, primarily because the smallest size of the flaw in the tested specimens is larger than the critical flaw size (see [20]). The discriminant here seems to be the grade of crack blunting: the edge-cracked specimens, which fail at smaller stretches, are well approximated by the 'pure' LEFM model, where the ultimate stretch decreases with the square root of $a$. On the contrary, the plates containing central cracks (specimens CC1–CC4) resist to larger stretches and do not agree well with the power law trends, suggesting that crack blunting plays a fundamental role.

To the latter group of specimens, we have applied the 'modified' model previously described. The results are summarized in Figure 13, where the quantity $K_t(\lambda_U - 1)$ is plotted as a function of the crack length $a$. Note that the quantity $\frac{1}{E}(K_t(\lambda_U - 1)E)$ represents the normalized true stress at the notch root at incipient failure, and can be therefore considered a material property, independent of the presence of the flaw. From the observation of the results reported in Figure 13, it can be noticed that such a local stress seems to be an appropriate parameter for quantifying the material failure in the case of the centered-cracked plates CC1–CC4, but not in the others. In other words, where crack blunting allows rupture at larger stretches, failure can be predicted by a local stress quantity rather than by a fracture mechanics-related one.

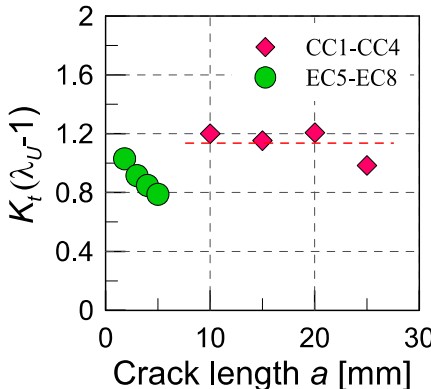

**Figure 13.** Normalized notch root true stress at incipient failure. The effect of crack blunting is enhanced for the specimens CC1–CC4, with the failure stress being approximately constant (dashed line).

## 4. Conclusions

The present paper investigates the defect tolerance capability of flawed silicone specimens subjected to tensile loading, adopting both a theoretical and an experimental approach. Silicone rubber has been chosen as illustrative of the typical response of soft matter in general, including other polymers, gels, and biological tissues. Different flaw geometries, sizes, and rates of the applied strain are examined in the experimental tests, with the aim of understanding their influence on the macroscopic mechanical response of the samples. The presence of internal defects, in the form of microbubbles, is also considered. The experimental response is monitored by measuring the applied force and using the Digital Image Correlation (DIC) technique to obtain precise two-dimensional displacement and full-field strain maps.

Flawed samples of soft materials subjected to tensile loading undergo a remarkable notch blunting prior to failure, which tends to reduce the stress concentration due to the presence of the flaw. Moreover, when notches are contained in nonconfined thin elements, a further notch blunting occurs as consequence of the local buckling of the material in the compressed zones, which arises normally to the loading direction. Experimental tests on thin silicone plates with elliptical notches showed that, irrespective of the initial size of the notch, very high remote stresses are supported thanks to the favorable notch profile evolution under load. With respect to the strain rate effects, we can observe that the flaw sensitivity is augmented at higher rates while, if a sufficiently slow strain rate is applied, the ultimate strain before failure is less affected by the size of the initial flaw. Indeed, slow deformation rates allow the internal microstructure of the material in high strained regions to be rearranged and to fail locally, with the consequence to smooth out the peak strain arising close to the geometric discontinuities.

A simple analytical model is proposed to account for the effect of crack blunting, in terms of the increase of the tip radius with the remote applied stretch. Although based on the assumption of linear elastic behavior, the application of the model to the experimental results has provided an interesting insight into the defect tolerance of cracked thin plates under tensile loading. Specifically, we have observed a transition from a typical flaw-size-dependent failure, as predicted by linear elastic fracture mechanics, to rupture occurring at a constant theoretical strength of the material. Our results show that, even at flaw lengths larger than the critical size found in other studies [20], silicone rubber specimens can withstand large stretches thanks to the flaw reshaping allowed by their peculiar microstructure. We are confident that the obtained results might be applied profitably to the evaluation of the safety levels of notched soft structural components, commonly found in numerous advanced applications.

**Author Contributions:** Conceptualization, A.S.; Data curation, R.B. and F.A.; Methodology, R.B.; Supervision, A.S.; Visualization, M.T.; Writing–original draft, M.T.; Writing–review & editing, A.C.

**Funding:** This research received no external funding.

**Conflicts of Interest:** The authors declare no conflict of interest.

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
