# Peer review of "How Soft Polymers Cope with Cracks and Notches"

_applsci, doi:10.3390/app9061086_

Author Response

The manuscript experimentally study the fracture in rubber like materials Sylgard 184. Failure of soft materials is a classical and hot topic now; and the current manuscript is a timely and relevant study. The manuscript is well-written and can be published in Ap- plied Sciences. However, the current manuscript still has some loose ends and room for improvement. I have some comments for the authors to consider:

(1)   The basic idea of flaw sensitivity and flaw insensitivity is as follows […]

We are grateful to the Reviewer for the valuable suggestions. In the revised manuscript, we have included the suggested references and better specified the purpose of our study. In particular, an explanation of the procedure we have adopted to assess the defect tolerance of flawed soft polymeric sheets has been added in the Introduction, in the Discussion of the results (Sec.3.3) and in the Conclusions.

(2)   Many relevant literature on fracture of soft materials are not cited. For example […]

Some of the references are now cited in the Introduction.

(3)   Another point about the adopted model. The region ahead of […]

The reason for we have used the neo-Hookean model is precisely that remarked by the Reviewer, that is, that at low to moderate stretches the model agrees well with the experimental curves. Incidentally, this is the range of stretches of most of our experiments. An explanatory sentence has been added in Sec.2 of the revised manuscript.

Reviewer 2 Report

Eq. (9) in the pdf manuscript was not displayed properly. 

The authors should discuss the limitations of the Digital Image Correlation technique, e.g., the smallest detectable flaw size and shape. 

Author Response

Eq. (9) in the pdf manuscript was not displayed properly.

The equation format has been corrected.

The authors should discuss the limitations of the Digital Image Correlation technique, e.g., the smallest detectable flaw size and shape.

Several parameters affect the reliability and accuracy of the DIC technique. Recent studies have shown the potential of the method for the measurement of deformation at the microscale, with resolution in the range of nanometres. In comparison, our experiments have much coarser resolution requirements (in the scale of the millimetre). A comment on the parameters which might influence the DIC results has been added in Sec.2.

Reviewer 3 Report

Dear Authors, the results presented in this manuscript are a summary of those presented and discussed in other three papers published by the same Authors.

The contribution to knowledge and the novelty is not clear to the reviewer.

I would suggest that the manuscript highlights better the difference with the current literature.

Author Response

Dear Authors, the results presented in this manuscript are a summary of those presented and discussed in other three papers published by the same Authors.

The contribution to knowledge and the novelty is not clear to the reviewer.

I would suggest that the manuscript highlights better the difference with the current literature.

The purpose of the present work is to collect the results of several experiments and provide a unified interpretation, in particular, by considering the defect tolerance in relation to the blunting occurring at the tip of cracks and notches under the ultimate remote stretch. In order to better clarify this point, we have added some relevant comments to the Introduction and the Conclusions sections.

Round  2

Reviewer 1 Report

The version is good for acceptance.